# Testing and Evaluation of Low-Cost Sensors for Developing Open Smart Campus Systems Based on IoT

**DOI:** 10.3390/s23208652

**Published:** 2023-10-23

**Authors:** Pascal Neis, Dominik Warch, Max Hoppe

**Affiliations:** School of Technology, Department of Geoinformatics and Surveying, Mainz University of Applied Sciences, 55128 Mainz, Germany; dominik.visca@hs-mainz.de (D.W.);

**Keywords:** IoT, low-cost sensors, smart campus system, open source, occupancy estimation

## Abstract

Urbanization has led to the need for the intelligent management of various urban challenges, from traffic to energy. In this context, smart campuses and buildings emerge as microcosms of smart cities, offering both opportunities and challenges in technology and communication integration. This study sets itself apart by prioritizing sustainable, adaptable, and reusable solutions through an open-source framework and open data protocols. We utilized the Internet of Things (IoT) and cost-effective sensors to capture real-time data for three different use cases: real-time monitoring of visitor counts, room and parking occupancy, and the collection of environment and climate data. Our analysis revealed that the implementation of the utilized hardware and software combination significantly improved the implementation of open smart campus systems, providing a usable visitor information system for students. Moreover, our focus on data privacy and technological versatility offers valuable insights into real-world applicability and limitations. This study contributes a novel framework that not only drives technological advancements but is also readily adaptable, improvable, and reusable across diverse settings, thereby showcasing the untapped potential of smart, sustainable systems.

## 1. Introduction

The process of urbanization leads to significant population concentrations in expanding metropolitan areas. This requires the intelligent management of issues such as traffic and mobility, pollution, energy, waste, and security. Given this background, a smart campus or a smart building can thus be conceptualized as a small-scale version or microcosms of smart cities, which bring both new opportunities and challenges in terms of technology and communication integration [1]. However, this is only a context-related umbrella term.

In a comprehensive literature review, [2] did not find a universal definition for a smart campus, and therefore drew attention to the specific contextual nature of this concept. As such, although developments must be interpreted with a keen understanding of the unique needs, resources, and circumstances of each setting, intelligent systems should, in our understanding, be designed and implemented with sustainable transferability in mind.

Smart management systems are particularly relevant for public buildings. These institutions not only have a responsibility to exemplify sustainable development, but they must also exert a positive impact on the cities and regions in which they are situated. In recent years, the digitalization trend has accelerated the importance of such smart building systems. On a campus, buildings are part of the physical infrastructure. Among other things, the systems aim to improve or simplify building operations and user experience through the integration and use of the Internet of Things (IoT) and its connected sensors [3,4].

In this context, information such as real-time crowd counts, movement flows, or air quality at indoor or outdoor events has gained unprecedented relevance due to the COVID-19 pandemic. In order to enforce and comply with hygiene protocols, it will be necessary to implement measures that further monitor current visitor numbers and movements inside and outside buildings [5,6,7,8,9].

In particular, the application of low-cost sensor technology can facilitate the cost-effective collection of diverse data and information. This includes climatic data such as temperature, air quality, or humidity [10,11], as well as people, vehicle, and environmental data such as light or noise levels [12,13]. Some of the most common applications for smart building or campus systems include visitor, space, energy, or parking management [4,14,15]. Such systems provide operators with enhanced event-scheduling capabilities and help manage resources. For visitors, these potential systems provide relevant information about location, available resources, and facility occupancy.

However, the development of these smart building or campus systems is not without challenges, including privacy issues, security risks, and technical difficulties [16,17]. For larger building complexes, monitoring visitors at building entrances is insufficient to provide meaningful information about visitor distribution. The implementation of comprehensive camera surveillance is often considered inefficient due to high technical requirements, acceptance issues, and potential violations of personal rights [18,19].

In this paper, we demonstrate the prototyping of a smart campus system using IoT based on low-cost sensor technology, open source software, and openly documented hardware components. Unlike other studies that generally focus on detecting and measuring specific variables for a smart building or campus system [14,20,21], we test and evaluate various methods to represent different use cases. These include real-time monitoring of visitor counts, room and parking occupancy, and collection of environmental and climate data. We prioritize sustainability and reusability of the overall system, focusing on open and standardized communication using established standards and interfaces from the geoinformatics domain. This approach facilitates data provision and the visualization of the collected data by the subsystems.

## 2. Related Work

Sensors for capturing different types of information are utilized across numerous areas. The internet-based access and provision of this data consequently create a network with a wide range of sensors. The emerging IoT thereby provides an architecture and infrastructure that enables many applications across various areas, such as Smart City Environment [3], Smart Home [22], Smart Campus [23], and Smart Traffic Systems [24]. Characteristics of the IoT architecture include standardized interfaces and protocols, as well as being open, scalable, and flexible. It can generally consist of an application, network/service, and sensor layer [25]. The integration and use of artificial intelligence in this context is becoming increasingly significant [12,16,26].

Overall there exist various methods to determine the real-time count of individuals or visitors at a specific location within a building (e.g., entrance), room, or area (e.g., restroom, hallway) using IoT. At its simplest, PIR sensors (passive infrared light) or motion detectors can be employed for room occupancy or person counting [27,28]. This requires enough motion detectors and is often used in a basic configuration to detect activity or identify room occupancy. In addition, there exist high-precision infrared array sensors which, depending on the surrounding temperature, can provide an accuracy of up to 90% in visitor counting [29]. Bluetooth (BLE) and Wireless Local Area Network (WLAN) technologies or networks are often employed to determine visitor count or room/building occupancy [30,31,32]. Here, data packets (broadcast messages) sent by smartphones, laptops, tablets, and wearables are used. The received signal strength indicator (RSSI), which plays a vital role in the captured communication of surrounding devices, can infer the distance between the sender and receiver. The authors of [33] have demonstrated that this is in principle useful for maintaining safety distances. In combination, BLE and WLAN can also be used to locate individuals [34,35,36]. In addition, camera-based detection of individuals and the subsequent counting have already confirmed good-to-excellent results in many demonstrated applications, with an accuracy of over 90% [37,38]. In the studies, cameras were either placed above a door or passageway in a top-down view [39] or in a slightly tilted position for better detection of the head and shoulder area [40,41]. In these latter studies, computer vision-based approaches are typically used to recognize people, track them, and thereby determine visitors. The methods rely on neural networks, blob detection, or Histograms of Oriented Gradients (HOG) and Local Binary Patterns (LBPs) for person detection [40]. The accuracy of these systems may strongly depend on the amount of training of the neural network and the selection of the features [42]. In addition to self-created and trained models, object detection systems or algorithms such as the “You Only Look Once” (YOLO) algorithm [43] or further optimized approaches like YOLO-based People Counting [44] are suitable. Furthermore, it is possible to detect individuals using Time-of-Flight (ToF) sensors [45], and to track them [46]. Variants using a sensor array system integrated into the floor (carpet) are also possible [47].

Additionally, the application of IoT within the context of individual transportation is not a new concept. It can be utilized for smart parking within buildings [48], in smart traffic management Systems [49], in the field of autonomous driving [50], or for monitoring the current traffic volume [51]. Different sensors are suitable for measuring traffic. These may include point magnetic sensors or induction loops in the roadway [52], GPS sensors and Radio Frequency Identification (RFID) tags in vehicles [53], or camera-based methods, for instance, for controlling traffic lights at an intersection [54], or for measuring the flow of traffic. The authors of [55] demonstrated what a computer vision system for real-time traffic measurement might look like. In addition to the approach of capturing traffic based on blob or contour recognition, they also examined a feature-based approach. The latter was also employed by [56] to identify vehicles using the OpenCV library and thereby determine the traffic volume. OpenCV is a free and extensive library in the field of computer vision and machine learning [57]. There are several studies that demonstrate its use in measuring traffic flow of vehicles [56,58,59], bicycles, or people [60,61,62]. For object detection to determine traffic, the YOLO algorithm, which is available in various versions, is also often used [63]. Depending on the application, there are special versions for real-time detection of different road users [62,64,65,66].

## 3. Towards an Open Smart Campus Monitoring and Information System

The developed Open Smart Campus System consists of various components. has been developed as a prototype at the Mainz University of Applied Sciences campus and put into operation for a test period [67]. The university has a total of four locations, with the prototype presented here implemented and tested on the main campus site.

### 3.1. Architecture of the Prototype System

The system basically consists of three components: Client, Web Service(s), and Sensors. Figure 1 shows the architecture of the prototype with its individual components. In addition to the various sensors (left) for each use case, it also includes the controller layer of the building’s Wi-Fi of the Mainz University of Applied Sciences, which can provide information about connected and logged-in devices per access point. The transmission of the readings from the individual sensors and their data storage is carried out via the Open Geospatial Consortium (OGC) SensorThings API [68] (center of Figure 1). The Fraunhofer IOSB’s freely available FROST server is used as an implementation of the standard, connecting the various sensors in a star network topology to the central processing server [69]. A web dashboard displays the measured values and provides access to additional analysis results and forecasts for different user groups (e.g., university management or students navigating the building). The background maps of the buildings, floors, or rooms required in the web dashboard are integrated in accordance with the OGC Web Map Service [70] and OGC Web Feature Service [71] specifications. The freely available GeoServer [72] from the nonprofit Open Source Geospatial Foundation organization (OSGeo) is used as the implementation. The geodata used are partly taken from the OpenStreetMap project [73] and have also been extended by the floor plans of the Mainz University of Applied Sciences.

The information system is also designed to generate the information necessary to make future predictions about the number and distribution of visitors, the flow of people, and the use of space and parking. The model applied for this purpose was presented by [74] and is expected to be implemented in a forecast module in the future. The intention is to establish a process via the OGC Web Processing Specification [75] that recognizes patterns from the input data using trained machine learning methods to infer from the current building occupancy to the future building utilization.

### 3.2. Low-Cost Sensors for the Use Cases

Depending on the implementation of the measurement method for each of the mentioned use cases in this article, the use of Single-Board Computers (SBC) such as Raspberry Pi Zero 2, Model 3, or 4 is suitable, as in other studies [14,19,76], due to reasonable hardware costs. The approximate cost for setting up a single room with the proposed sensor technology is around $100 to $150, depending on the specific sensors and SBC models chosen. This is significantly less expensive than commercial smart building solutions, which can cost upwards of $1000 per room for similar functionalities. 

To provide a comprehensive overview of the sensors used in this study, Table 1 summarizes their key specifications [77,78,79].

As shown in Table 1, these SBCs are already equipped with various low-cost sensors or can be extended with these through USB, a Camera Serial Interface (CSI), or a General-Purpose-Input/Output interface (GPIO). In this study, a Raspberry Pi Zero 2 or a Raspberry Pi 4 Model B with 8 GB RAM was used with its pre-installed WLAN and BLE modules and, if necessary, an additional WLAN module as sensors. A Raspberry Pi Camera V2 and a Tapo C110 home security WLAN camera were also utilized. To measure environmental and climate information, a DHT11/DHT22 was used for temperature/humidity, an MQ-2 for gas/smoke, and an MH-Z19 sensor for CO2. As an alternative, a Raspberry Pi HAT like the Enviro+ was used. This allows the measurement of air quality (harmful gases and particles), temperature, pressure, humidity, light and noise levels. 

To implement the use case of estimating the number of people and room occupancy in the Smart Campus System, different methods were tested and evaluated. When a smartphone, smartwatch, laptop, tablet, etc. with a BLE interface is turned on, the device periodically sends broadcast messages to all nearby devices. Bluetooth has a theoretical range of up to 100 m outdoors and up to 20 m inside a building with walls. If the broadcast messages sent in the environment are collected by an SBC with a Bluetooth interface using discoverable mode, relevant metadata such as MAC address and RSSI can be further analyzed, and conclusions about at least the number of devices in the vicinity can be calculated. However, the number of devices does not necessarily correlate with the number of people, as they may be carrying multiple devices.

Similarly, modern electronic devices have a WLAN interface. If it is turned on, broadcast messages are sent periodically. To be able to receive such data packets, a WLAN module must support the so-called monitor mode. The Raspberry Pi’s built-in WLAN module supports this, but the operating system (Raspberry Pi OS) does not. For this reason, a second WLAN module is used via the USB interface, and the built-in WLAN module comes into action for communication with the data storage layer. The range of WLAN devices can ideally be several hundred meters outdoors and between 30 and 40 m inside a building, although the range is always dependent on the hardware and frequency used. Aircrack was used to continuously record the data traffic and the associated devices in the vicinity with their MAC address and their RSSI as a so-called packet sniffer. In addition, as mentioned, the existing WLAN on campus was used. At Mainz University of Applied Sciences, just over 180 access points are available for indoor and outdoor use. The access points used are Aruba7030 Mobility Controllers [80]. This model provides an API, which, among other things, returns the devices in the vicinity (connected and unconnected).

Time-of-Flight (ToF) sensors can be used to measure distances using the runtime method, which can also be used to estimate the number of people passing by an installed sensor. Depending on the sensor used, different numbers of pixels are available. In the development of the system presented here, the TeraRanger Evo Mini from TeraBee was used. This can measure distances between 3 and 330 cm with a field of view of 27 degrees [81]. The sensor can be configured in either a single or multi-pixel and short or long measurement mode. It was connected to the SBC via USB. In addition to the Evo Mini, the Evo People Counter from TeraBee was also examined. The shape and technical specifications of the Evo People Counter is identical to the Evo Mini. Figure 2 shows the EvoMini and Evo People Counter sensor (a), and the mounting options of the sensor from above (b) or on a wall (c). The Evo People Counter offers a direct interface for measuring or retrieving the recorded visitor frequency from the sensor for all common operating systems. A potential detection range of 5 to 150 cm in the area of a door or corridor is specified for this [82]. A custom Python 2.7 script is used as software. With the Evo Mini, an algorithm must be implemented to detect the movement and direction of a person from the measured distances. However, the manufacturer offers some examples on GitHub that simplify the implementation [83].

As mentioned, image processing can be used to detect objects such as people or vehicles and track their movement using approaches with neural networks, blob detection methods, HOG-LBP, or YOLO. Both are applied in the use case for determining the number of visitors and parking space occupancy. For this study, two low-cost cameras were selected. The Raspberry Pi camera module 8MP v2 can be connected directly to the SBC via the existing CSI. For static images, a resolution of 3280 × 2464 pixels is possible. Videos are supported in 1080 p with 30 frames per second (FPS) while the field of view is 62 degrees. The Tapo C110 home security Wi-Fi camera was used as a second camera. Via a Wi-Fi connection, it delivers a video stream with a resolution of up to 2304 × 1296 pixels at up to 30 FPS and a field of view of 105 degrees. A self-implemented Python script is used as software, utilizing OpenCV [84] and a MobileNet Single Shot Detector (SSD) or the YOLO method for deep learning-based object detection. The object tracking required for determining visitor frequency and measuring parking space entry/exit or occupancy is centroid based. Comparable approaches can be found on GitHub and other sites [85].

The collection and storage of environmental and climate information is significantly simpler compared to the methods previously mentioned. Using available Python scripts, readings are retrieved for the desired sensors and processed accordingly. For example, they can be sent to the OGC SensorThings API used for storage or, if desired, they can be processed for direct verification and possible notification if a threshold is exceeded.

Although the system is designed for efficient data transmission, the possibility of packet loss during communication between the sensors and devices cannot be entirely ruled out. Factors such as network congestion, interference, and signal degradation could potentially result in packet loss. Particularly, BLE operates in the often congested 2.4 GHz frequency band. This band is shared by numerous devices and technologies, leading to potential interference and collisions. As highlighted in previous experiments [86,87,88], BLE and Wi-Fi can experience challenges in providing real-time and reliable service for time-critical applications due to packet loss and collisions, especially when multiple connections are involved. Such issues can be exacerbated in scenarios with numerous devices operating simultaneously, leading to increased chances of packet collisions and subsequent data loss. However, for our specific application, real-time transmission is not a primary concern. Our system is designed to use data points in 5 min time windows, which increases the time frame for successful data transmission. This design choice inherently reduces the impact of transient packet loss or temporary network congestion. By spacing out our data transmissions, we can ensure a higher degree of reliability and reduce the potential for packet collisions, making the system more resilient to the challenges commonly associated with BLE in crowded frequency bands. Furthermore, the system employs a retransmission mechanism, where the sensors will attempt to resend the data packet if an acknowledgement is not received within a specified time frame. Additionally, the system handles occasional packet loss gracefully, ensuring that minor disruptions do not significantly impact the overall performance and reliability by interpolating single missing data points.

### 3.3. Web Dashboard

The measurements collected by the various sensors and their analysis results are combined in a web dashboard for the users (e.g., university management or students navigating the building). This dashboard is built on a Single Page Application (SPA) model using JavaScript, utilizing the React library for better user interface and responsiveness and Leaflet as a central map component. Custom Webpack configurations allow for a modular design, making the dashboard adaptable for both development and production environments. Real-time data updates are facilitated through the WebSocket protocol to fetch current data from the central FROST server, which gathers the readings of the various connected sensors on the SBC [89,90,91]. The basis for visualization is a map with the rooms of the Mainz University of Applied Sciences campus, or more precisely, Its main building, which consists of almost 450 rooms (including offices, toilets, staircases, storage rooms, etc.) spread over four floors. The focus of the developed Smart Campus System was on the approx. 50 lecture halls, 8 PC pools, and 7 labs at the location. Figure 3 shows the main building of the University and the official parking spaces for students, staff, and visitors. The individual colored points visualize the main entrance, the cafeteria, the location of the camera for parking management, and the lecture halls used in the test and evaluation phase.

Unlike existing systems, the developed dashboard can display not only static but also real-time and historical information for the rooms. In addition to the number of people currently in the buildings (or floors or rooms), alternative visualizations of flows or hot spots of people in the building are also possible. Students, faculty, and visitors can thus obtain information about building or room occupancy in advance or, if necessary, be automatically notified in case of exceeding the maximum number of people or other events, provided integration with the existing systems is in place. Figure 4 shows the developed web dashboard with the map and a color scale of room occupancy on campus. In addition, the current and past occupancy for the selected room is illustrated in detail as an example.

## 4. Low-Cost Sensors Used: Insights and Experiments

The various field tests on the measurement procedures of the presented use cases took place in the context of seven different courses, each in the summer semester 2022 and winter semester 2022/2023. The courses were held in the form of lectures, exercises, exams, and colloquia at the Department of Engineering and the Department of Geoinformatics and Surveying in different semesters and weeks of the lecture period. Group sizes ranged from 12 to 40 students, and the subjects were informed about the experiments and trials, but not about the exact measurement procedures until later.

Figure 5 provides a visual representation of the devices utilized in our study. On the left are the TeraRanger Evo Mini sensors connected to a Raspberry Pi 4 in a protective case. Adjacent, on the right, is a power bank ensuring flexible placement and uninterrupted power supply to the sensors and emphasizing the system’s mobility and adaptability. On the upper right stands the Tapo C110 camera, and on the bottom right are two Raspberry Pi 4 devices, one equipped with the Enviro+ sensor suite and the other one with the PiCamera V2. All three Raspberry Pis are also connected to a separate Wi-Fi dongle, enabling the parallel use of the monitor mode while transferring data.

Visitors to a lecture hall were counted as they entered, stayed, and left. The lecture halls used had the rectangular floor plan shown in Figure 6 and always had an entrance door. The SBC with the WLAN & BLE module and the environmental climate sensors was positioned in the center of the room (see Figure 6a). The camera facing into the lecture room was mounted above the door and the ToF sensor near the door frame (see Figure 6b,c).

As confirmed by many published studies, the image/video-based sensor yielded the best results through the evaluation of the camera with an average accuracy of over 90% for the events tested. The worst results were obtained with the ToF sensors. Here, the number of visitors could only be determined with an average accuracy of 40%. In our first experiments, interesting measurements were also obtained with the WLAN and BLE sensors. The captured BLE devices and the number of people had a correlation coefficient r of 0.82. Compared to previous published studies, this value has a significant effect for BLE devices; Ref. [92] concluded in their experiments using BLE that only 11% of people had a BLE device with them. Furthermore, Ref. [13] also mention that the number of BLE devices would be negligible compared to WLAN devices when counting people in the vicinity. This could not be confirmed in the tests conducted here. Figure 7 shows the number of detected WLAN and BLE devices and the manually counted number of people in the respective event room. In Figure 7, for the study of WLAN devices, two outliers at 55 and 70 people can be seen. These two measurements come from a kick-off event in the first week of the lecture period.

Due to the insufficient quality of the results of the ToF sensors for use in a visitor information system, this method was not pursued further. The approach to measure visitor frequency through image evaluation on a large scale is concerning from the perspective of data privacy and was therefore also not pursued further in the use case at the Mainz University of Applied Sciences campus.

For the use case of parking lot management with information about the number of occupied and free parking spaces, a camera was installed around the entrance to the parking lot, as shown in Figure 3. The ongoing traffic in and out of the parking lot area was evaluated using a script via the live stream that is thus available (see Figure 8). The line marked in yellow in the image is used to distinguish the direction of travel.

## 5. Challenges and Lessons Learned

The results in the various testing of the respective use cases show that the accuracy of the results can vary depending on the selected method and the choice of sensors. For the determination or estimation of the number of visitors or the frequency of visitors, the used ToF sensors required special care when being mounted and aligned. The manufacturer provides appropriate recommendations for this. Likewise, the manufacturer also points out that the sensors can only count individual people. If several people pass the sensor close together, they are counted as one visitor. The testing in the lecture rooms showed that students often walked past the sensor in small groups, which is why there were differences in the number of counted and actual visitors. Therefore, if there are less frequented passages or areas, the investigated ToF sensor might also be suitable.

The good or very good results in visitor counting using the camera-created images or video stream evaluation were already known due to numerous studies. However, the hardware and software used in this article underpin the fact that corresponding results can also be achieved with low-cost cameras and open-source software. As already mentioned in this and other studies, complete video surveillance is, however, a cause for concern from a data privacy point of view and is not sought after in the information system at the campus in Mainz.

The use of BLE and WLAN sensors again showed that these two sensors or methods are well suited for detecting the number of devices, estimating the number of people by correlation, and using them in an information system, at least in comparison to other studies. In the approach shown here, the threshold value for filtering the detected RSSI values had to be adjusted individually for both methods, depending on the room, because otherwise, too many devices would be found and, thus, no meaningful correlation between detected devices in the environment and people could be calculated. Rooms around the cafeteria or the main entrance, for example, posed greater difficulties (see Figure 3) because of the high flow of visitors. Nevertheless, there is further potential; for example, the number of packets exchanged during communication between the sensor and devices in the environment could also be recorded and considered when determining the number of visitors. From a privacy point of view, however, this should be done directly in the sensor itself and without persistence of this data.

There is also a difference between indoor and outdoor use of BLE and WLAN signals. The signals behave differently due to walls and materials in the environment, although the tests in this article only took place in lecture rooms and thus, indoors. Isolated outdoor tests nevertheless showed tendencies that results with similar accuracy can be achieved, assuming an adequate calibration as described.

In the parking management use case, vehicles were detected via the YOLO algorithm, which was also used in further processing. In various preliminary tests, YOLO was one of the fastest in object detection compared to other approaches. However, it is possible that another approach is better suited for the evaluation of video streams from web cameras, for example. This should be examined depending on the image source and format. Another issue that has not yet been investigated is the influence of weather. Our experiments took place in summer, which means that difficult lighting conditions, for example, played a minor role for the time periods relevant to us.

## 6. Conclusions and Future Work

To ensure the sustainability of the presented smart campus system and components from a technological perspective, the implementation was carried out using open-source software, open data, and open standards and data formats. The result is a prototype for a modular and open-source system that can be reused and adapted by other universities, small and medium-sized enterprises, or public institutions. Technological transferability is thus comparatively easily enabled, and parts such as the involved low-cost sensors can be used or individually adapted.

The prototypical information system we developed and applied on the campus of the Mainz University of Applied Sciences indicates that low-cost sensors can provide reliable data for real-time analytics and decision-making systems for various users, from students to facility managers. Specifically, the image/video-based sensor yielded an average accuracy of over 90% for visitor counting, whereas the BLE devices showed a significant correlation coefficient r of 0.82 with the number of people present. This is particularly significant given data privacy considerations included in our research, areas that are often not given due attention [93].

These metrics affirm that the examined low-cost sensors could provide good-to-very good results, as reflected in the consistent data obtained in multiple iterations. The intrinsic validity of these results is derived from comprehensive evaluations, as the experiments were conducted in seven different courses during both the summer and winter semesters of 2022 and 2022/2023. The choice of different academic settings and the heterogeneous nature of the group sizes, ranging from 12 to 40 participants, provided a diverse sample, facilitating more general and reliable inferences about the efficacy of the sensors. Nevertheless, it must be emphasized that the robustness and validity of these results can only be confirmed by comparative analyses with analogous experimental setups.

Furthermore, our research adds a new layer to the ongoing discussion about the increasing prevalence of sensors and IoT devices in the smart campus context. It points out the need for open systems that can be scaled up as technology evolves. In this context, the transition to an integrated digital building or campus will rely heavily on information and communication technology (ICT) and IoT infrastructures [94,95]. Depending on the structural conditions or the existing infrastructure, our study suggests that smart technologies, when integrated effectively into existent facilities, can contribute to improved services and decision-making processes [96,97,98].

For future research and as mentioned above regarding the results, it would be relevant to test the adaptability of our framework to different environments and conditions. Overall, the work adds a practical perspective to the current landscape of smart technology integration in public and educational settings, emphasizing the role of open-source systems and data privacy.

## Figures and Tables

**Figure 1 sensors-23-08652-f001:**
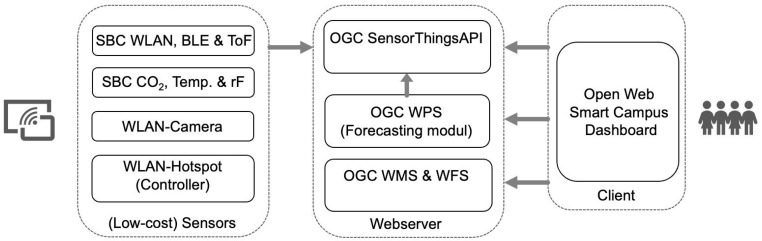
Architecture of the Open Smart Campus System.

**Figure 2 sensors-23-08652-f002:**
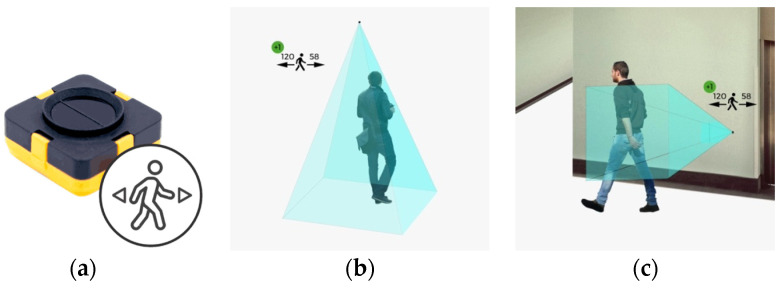
EvoMini and Evo People Counter Sensor (**a**): the mounting options of the sensor from above (**b**) or on a wall (**c**). Image source: [82].

**Figure 3 sensors-23-08652-f003:**
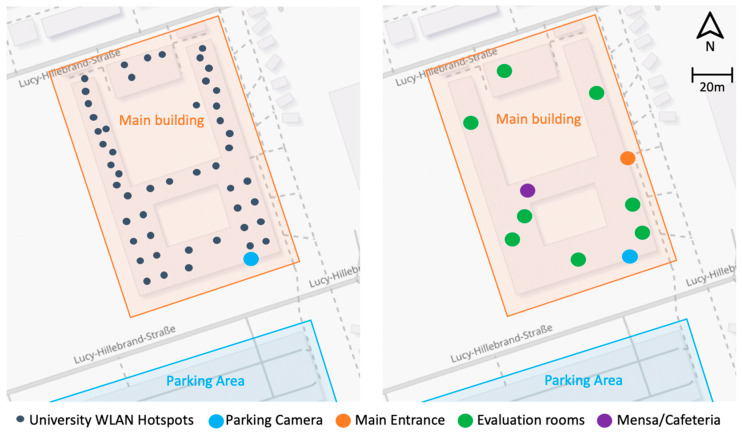
University map with main building and parking area. © Background map: OpenStreetMap Contributors.

**Figure 4 sensors-23-08652-f004:**
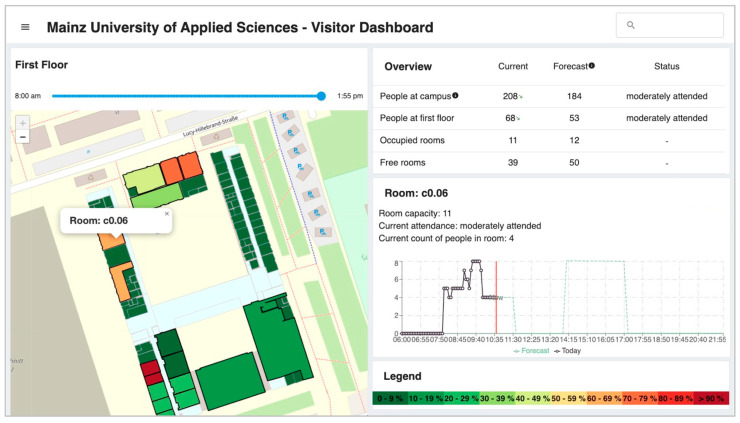
Web dashboard of the developed Open Smart Campus System. The red line represents the current time.

**Figure 5 sensors-23-08652-f005:**
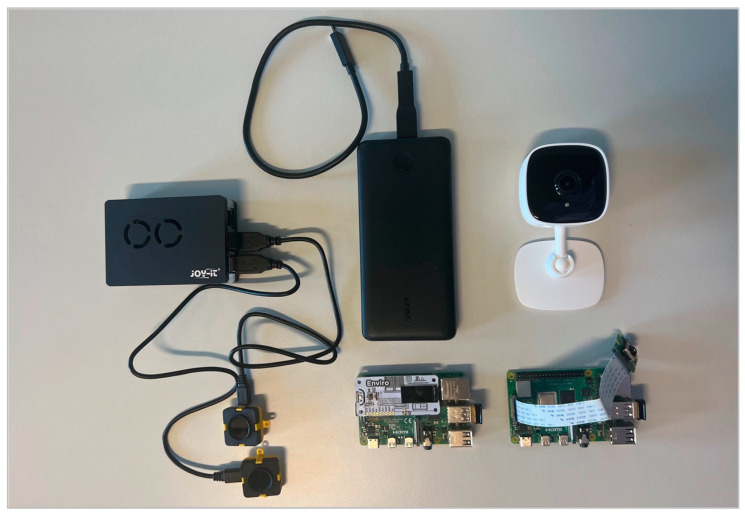
Showcasing of utilized sensors and devices.

**Figure 6 sensors-23-08652-f006:**
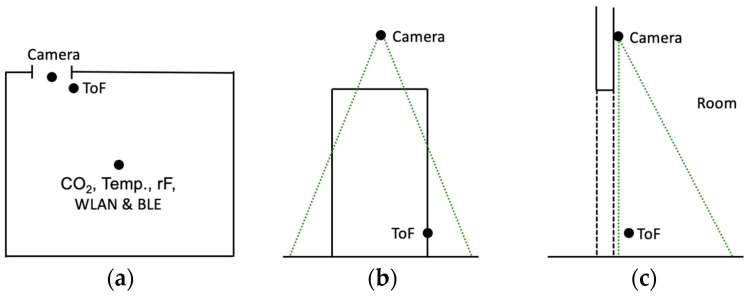
Floor plan of the lecture halls used and positions of the sensors: (**a**) Room layout; (**b**) door frontal view; and (**c**) door lateral view.

**Figure 7 sensors-23-08652-f007:**
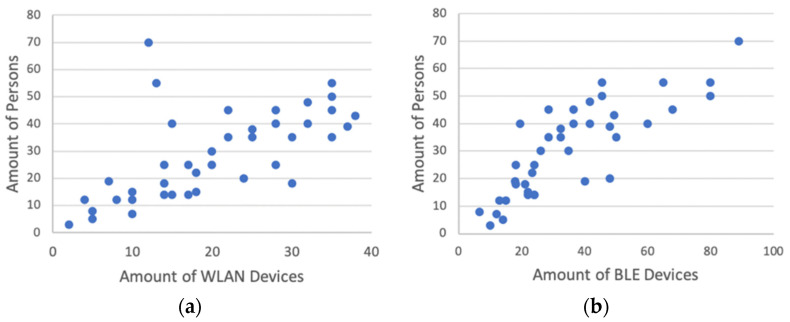
Number of detected (**a**) WLAN devices and (**b**) BLE devices compared to the manually counted number of people in the tests for the visitor frequency use case.

**Figure 8 sensors-23-08652-f008:**
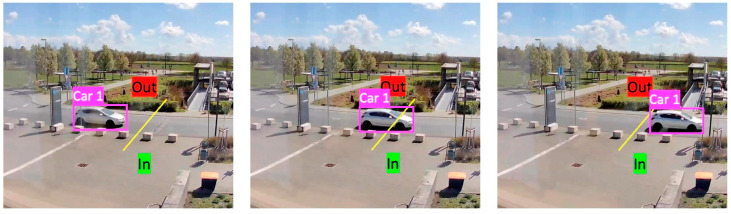
Detected vehicles in and out of the parking lot at the Mainz University of Applied Sciences campus for the parking management use case.

**Table 1 sensors-23-08652-t001:** Overview of the sensors and devices used.

Component	Functionality	Interface	Voltage	Power Consumption	Accuracy
BME280	Humidity and Pressure	GPIO/Enviro+	1.7–3.6 V	3.6 μA	±3%
DHT11	Temperature and Humidity	GPIO	3.3–5 V	2.5 mA	±2 °C
Evo Mini	Infrared ToF	USB	5 V	50 mA	±1.5 cm
Evo People Counter	Infrared ToF	USB	5 V	50 mA	±1.5 cm
LTR-559	Light and Proximity	GPIO/Enviro+	2.4–3.6 V	20 mA	N/A
MiCS-6814	Gas Sensor	GPIO/Enviro+	1.7–2.4 V	32 mA	N/A
PIR Sensor	Motion Detection	GPIO	3.3–5 V	0.8 W	N/A
Pi Camera V2	Camera	CSI	3.3 V	250 mA	8 MP
SPH0645LM4H-B	Digital Microphone	GPIO/Enviro+	1.6–3.6 V	600 μA	N/A
Tapo C110	Home Security Camera	WiFi	5 V	300 mA	3 MP
MQ-2	Gas/Smoke Detection	Analog/GPIO	5 V	N/A	N/A
MH-Z19	CO2 Measurement	UART/PWM	3.6–5.5 V	18–33 mA	±(50 ppm + 5%)

## Data Availability

The data presented in this study are available on request from the corresponding author.

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
