# Peer review of "Testing and Evaluation of Low-Cost Sensors for Developing Open Smart Campus Systems Based on IoT"

_sensors, 2023, doi:10.3390/s23208652_

Round 1
Reviewer 1 Report
i am grateful for the opportunity to review your manuscript.
i acknowledge and appreciate your intent, namely to distinguish your work from "other studies that focus on detecting and measuring specific variables" by instead prioritising "sustainability and reusability" interfacing with "the geoinformatics domain".
i recognise this to be the primary contribution of your work.
i offer the following for your consideration:
- in the second paragraph of the section 'Related work', "it is" is contracted to "it's". please attend to this and ensure that there be no similar contractions elsewhere in your manuscript.
- please re-express the heading for Section 4, for example, by inverting the word sequence (eg, "the low-cost sensors used")
- likewise, within the text of this same section, please re-express "data privacy-wise concerning" to (for example) "concerning from the perspective of data privacy".
- finally in this same section, i found the proportion of your analysis / discussion imbalanced between the use case in the parking lot, and the use case of the room-based sensors. at the very least, do insert a paragraph break between "... Sciences campus. For the use case of ..."; i do encourage you to develop your arguments from the parking lot use case further, within the manuscript. presently, the imbalance is very jarring. if you are limited by word-counts, then perhaps omit the parking lot use case altogether. the present imbalance is unsatisfactory.
Author Response
Dear Reviewer,
We highly appreciate the time and effort put into the detailed review. The changes are highlighted in the paper. Below, please find our responses to each of the points raised:
In the second paragraph of the section 'Related work', "it is" is contracted to "it's". please attend to this and ensure that there be no similar contractions elsewhere in your manuscript.
We have amended the contraction "it's" to "it is" in the second paragraph of the 'Related Work' section. We have also reviewed the rest of the manuscript to ensure that no similar contractions are present.
Please re-express the heading for Section 4, for example, by inverting the word sequence (eg, "the low-cost sensors used").
The heading for Section 4 has been revised as suggested. The new heading is now "Experiments and Insights on the Low-Cost Sensors Used."
Likewise, within the text of this same section, please re-express "data privacy-wise concerning" to (for example) "concerning from the perspective of data privacy".
The phrase "data privacy-wise concerning" in the text of Section 4 has been revised to "concerning from the perspective of data privacy."
Finally in this same section, I found the proportion of your analysis / discussion imbalanced between the use case in the parking lot, and the use case of the room-based sensors. at the very least, do insert a paragraph break between "... Sciences campus. For the use case of ..."; I do encourage you to develop your arguments from the parking lot use case further, within the manuscript. presently, the imbalance is very jarring. if you are limited by word-counts, then perhaps omit the parking lot use case altogether. the present imbalance is unsatisfactory.
We appreciate your feedback on the imbalance between the two use cases. We have inserted a paragraph break between the discussion of the Sciences campus and the parking lot use case, as suggested.
The manuscript has been revised thoroughly to address the valuable comments provided by the reviewer. We believe these changes have significantly improved the quality and clarity of the manuscript.
Sincerely
Reviewer 2 Report
sensors-2601207
Title: Testing and Evaluation of Low-cost Sensors for Developing Open Smart Campus Systems Based on IoT
Indeed, the manuscript is difficult to follow. There are some points that need to be clearly known.
-How do the authors develop a web dashboard? Which platform has been used to create it? How have Single-Board Computers (SBC) been interfaced with it?
- What do you mean by low-cost? What is the approximate cost of the proposed setup?
-It will be good to add a table showing different sensors (with their specifications) in this study.
-Including a flowchart of the software used in this study will be good.
-Please include the real picture of the experimental setup (showing sensors, Single-Board Computers (SBC), etc. ) used in this study.
-Conclusion: “the examined low-cost sensors could provide good to very good results for the mentioned…” Is there any parameter to confirm this statement (good to very good results)?
- I there any mesh or star network used in this work?
-Line 332 “, the number of packets exchanged during communication between the sensor and devices in the environment..” Is there any possibility of packet loss during communication between the sensor and devices?
-It is better to list a comparison table to compare results with previous work.
-The novelty of the work should be clearly highlighted (in the abstract as well as in the conclusions).
Minor editing of English language required.
Author Response
Dear Reviewer,
We highly appreciate the time and effort put into the detailed reviews. The changes are highlighted in the paper. Below, please find our responses to each of the points raised:
How do the authors develop a web dashboard? Which platform has been used to create it? How have Single-Board Computers (SBC) been interfaced with it?
The manuscript has been revised to include a subsection detailing the development of the web dashboard.
What do you mean by low-cost? What is the approximate cost of the proposed setup?
We have added a paragraph to clarify the approximate cost of the proposed setup.
It will be good to add a table showing different sensors (with their specifications) in this study.
As suggested, we have included a table listing the specifications of different sensors used in the study.
Including a flowchart of the software used in this study will be good.
Figure 1 already contains all the individual hardware and software components and their respective relationships to each other. After consultation, we have come to the conclusion that this figure should be sufficient.
Please include the real picture of the experimental setup (showing sensors, Single-Board Computers (SBC), etc.) used in this study.
Since none of the authors are currently on site at the university, we are unable to provide an image at this time.
We would - as far as the editor allows - add it at a later stage or in a possible second round of review.
Conclusion: “the examined low-cost sensors could provide good to very good results for the mentioned…” Is there any parameter to confirm this statement (good to very good results)?
We revised the conclusion to include specific parameters that support our statement that "the examined low-cost sensors could provide good to very good results."
I there any mesh or star network used in this work?
We have clarified that a star network was used in this work.
Line 332 “, the number of packets exchanged during communication between the sensor and devices in the environment.” Is there any possibility of packet loss during communication between the sensor and devices?
The manuscript now includes information of the possibility of packet loss during communication between the sensor and devices.
It is better to list a comparison table to compare results with previous work.
In general, we agree with your comment, but in the specific case of this study, we feel it is not as necessary: The aim was to show that it is possible to achieve similarly high-quality results with a much cheaper, more sustainable and more flexible setup. In our opinion, this has been sufficiently demonstrated by the additional clarification at the end, which you also suggested.
The novelty of the work should be clearly highlighted (in the abstract as well as in the conclusions).
We have made it a point to explicitly highlight the novelty of the work in both the abstract and conclusions.
The manuscript has been revised thoroughly to address the valuable comments provided by the reviewer. We believe these changes have significantly improved the quality and clarity of the manuscript.
Sincerely
Round 2
Reviewer 2 Report
sensors-2601207
Title: Testing and Evaluation of Low-cost Sensors for Developing Open Smart Campus Systems Based on IoT
Some comments are not adequately addressed. Please revise the manuscripts as per the comments. Also, include the response in the manuscript and the response to reviewers.
-Please include the real picture of the experimental setup (showing sensors, Single-Board Computers (SBC), etc. ) used in this study. Address it properly; the Experimental setup must be shown in the manuscript.
-Conclusion: “the examined low-cost sensors could provide good to very good results for the mentioned…” Is there any parameter to confirm this statement (good to very good results)? Please discuss the parameters to confirm the statement (good to very good results).
-Line 332 “, the number of packets exchanged during communication between the sensor and devices in the environment..” Is there any possibility of packet loss during communication between the sensor and devices? Address it properly.
Minor editing of English language required
Author Response
Dear Reviewer,
Again we highly appreciate the time and effort you put into the review! The changes are highlighted in the paper. As before please find our responses to each of the points raised:
Please include the real picture of the experimental setup (showing sensors, Single-Board Computers (SBC), etc. ) used in this study. Address it properly; the Experimental setup must be shown in the manuscript.
We have included a photo of the devices and sensors used and an explanatory paragraph. For an exact understanding of the experiment setup, we refer to our sketches (Figure 6), which in our opinion is much more informative than a photo taken on site could be.
Conclusion: “the examined low-cost sensors could provide good to very good results for the mentioned…” Is there any parameter to confirm this statement (good to very good results)? Please discuss the parameters to confirm the statement (good to very good results).
We have revised the Conclusion again and, as requested, added a section that better contextualizes and discusses our results.
Line 332 “, the number of packets exchanged during communication between the sensor and devices in the environment..” Is there any possibility of packet loss during communication between the sensor and devices? Address it properly.
We have added to the section on packet loss that was added in the first round of reviews, mainly to provide technical details and our approaches to dealing with the problem of packet loss. In addition, we have supplemented other sources in this matter and hope to have discussed the topic thoroughly for our context.
Sincerely
Round 3
Reviewer 2 Report
sensors-2601207
Testing and Evaluation of Low-cost Sensors for Developing Open Smart Campus Systems Based on IoT
Thank you for allowing me to revise resubmitted manuscript titled "Testing and Evaluation of Low-cost Sensors for Developing Open Smart Campus Systems Based on IoT" I believe the submitted manuscript and presented work is suitable for publishing in the Sensors, except for some minor revision.
Minor revision:
Question:1
It will be good to add some more references related to Smart Campus Systems Based on IoT e.g., https://doi.org/10.3390/su142416630
Minor editing of English language required.
Author Response
Dear reviewer,
Thank you for the comments and especially for the concluding assessment!
We have added a total of four more references (including the one you suggested) at the appropriate positions to further support the topic of the Smart Campus system!
Best regards,
The authors